# Resting natural killer cells promote the progress of colon cancer liver metastasis by elevating tumor-derived stem cell factor

Chenchen Mao[1,2†], Yanyu Chen[2,3†], Dong Xing[2†], Teming Zhang[2†], Yangxuan Lin[4], Cong Long[2], Jiaye Yu[2], Yunhui Luo[2], Tao Ming[2], Wangkai Xie[2], Zheng Han[2], Dianfeng Mei[2], Dan Xiang[2]*, Mingdong Lu[1]*, Xian Shen[5]*, Xiangyang Xue[1,2]*

[1]Department of General Surgery, The Second Affiliated Hospital and Yuying Children's Hospital of Wenzhou Medical University, Wenzhou, China; [2]Department of Microbiology and Immunology, Institute of Molecular Virology and Immunology, School of Basic Medical Sciences, Wenzhou Medical University, Wenzhou, China; [3]Department of Pediatric Thoracic Surgery, The Second Affiliated Hospital and Yuying Children's Hospital of Wenzhou Medical University, Wenzhou, China; [4]Department of Thoracic Surgery, The First Affiliated Hospital of Wenzhou Medical University, Wenzhou, China; [5]Department of General Surgery, The First Affiliated Hospital of Wenzhou Medical University, Wenzhou, China

*For correspondence:
xiangdan@wmu.edu.cn (DX);
lumd@wmu.edu.cn (ML);
shenxian@wmu.edu.cn (XS);
wzxxy@wmu.edu.cn (XX)

†These authors contributed equally to this work

Competing interest: The authors declare that no competing interests exist.

**Abstract** The abundance and biological contribution of natural killer (NK) cells in cancer are controversial. Here, we aim to uncover clinical relevance and cellular roles of NK cells in colon cancer liver metastasis (CCLM). Here, we integrated single-cell RNA-sequencing, spatial transcriptomics (ST), and bulk RNA-sequencing datasets to investigate NK cells' biological properties and functions in the microenvironment of primary and liver metastatic tumors. Results were validated through an in vitro co-culture experiment based on bioinformatics analysis. Useing single-cell RNA-sequencing and ST, we mapped the immune cellular landscape of colon cancer and well-matched liver metastatic cancer. We discovered that GZMK+ resting NK cells increased significantly in tumor tissues and were enriched in the tumor regions of both diseases. After combining bulk RNA and clinical data, we observed that these NK cell subsets contributed to a worse prognosis. Meanwhile, KIR2DL4+ activated NK cells exhibited the opposite position and relevance. Pseudotime cell trajectory analysis revealed the evolution of activated to resting NK cells. In vitro experiments further confirmed that tumor-cell-co-cultured NK cells exhibited a decidual-like status, as evidenced by remarkable increasing CD9 expression. Functional experiments finally revealed that NK cells exhibited tumor-activating characteristics by promoting the dissociation of SCF (stem cell factor) on the tumor cells membrane depending on cell-to-cell interaction, as the supernatant of the co-culture system enhanced tumor progression. In summary, our findings revealed resting NK cells exhibited a clinical relevance with CCLM, which may be exploited for novel strategies to improve therapeutic outcomes for patients with CCLM.

## eLife assessment

This **useful** study draws on published single-cell and spatial transcriptomic data of colon cancer liver metastasis to clarify the pro- and anti-tumorigenic properties of NK cells. The authors discover increased GZMK+ resting NK cells in the tumor tissue and reduced abundance of KIR2DL4+

activated NK cells. However, the evidence is currently **incomplete**, as the models used to validate the hypothesis and claims are not adequate and lack the necessary controls.

## Introduction

Colon cancer is one of the most common malignancies, with more than 1 million new cases and 500 thousand deaths reported globally in 2020 (*Sung et al., 2021*). Approximately 19–26% of patients with colon cancer present with synchronous metastatic diseases at the first diagnosis (*Siegel et al., 2020*), among which 14.5–17.5% develop liver metastasis (*Siegel et al., 2020*; *Wang et al., 2023*). Although progress in the treatment of metastatic disease, including improved diagnosis and treatment strategies for liver metastases (*Chua et al., 2011*; *Zampino et al., 2016*; *Ruers et al., 2017*), increased cancer-directed surgery (*Wancata et al., 2016*), and the development of targeted therapies (*Piawah and Venook, 2019*), has greatly improved the survival of these patients in recent decades, the 2-year relative survival rate for patients diagnosed with distant-stage disease was 36% (*Siegel et al., 2020*), significantly lower than those without metastasis. Thus, colon cancer-derived liver metastasis (CCLM) is a clinical challenge that requires urgent development of novel treatment methods.

The tumor microenvironment (TME) is a dynamic environment that governs tumor behavior and is required for tumor cell survival, growth, proliferation, and metastasis (*Yao et al., 2023*). Unlike previous understanding, it has recently been proposed that certain immune cells present in the TME [such as myeloid-derived suppressor cells (*Liu et al., 2023*)], macrophages (*Ngambenjawong et al., 2017*), neutrophils (*Li et al., 2019*), and CD8[+] T cells (*Tiberti et al., 2022*) favor tumor progression. Thus, the contradictory findings of the pro- and anti-tumor effects of tumor-infiltrating immune cells require a better understanding of the immune features and profile heterogeneity of colon cancer liver metastasis (CCLM) for the future development of immune-modulatory strategies to stratify and target immune cells.

Natural killer (NK) cells, an important component of tumor-infiltrating immune cells, are cytotoxic lymphocytes that play a key role in recognizing and eliminating malignant cells and are therefore considered the early responders against tumors (*Maskalenko et al., 2022*). Among various immune cells, tumor-infiltrating NK cells are associated with prognosis in various cancers, such as colorectal cancer (CRC) (*Zhong et al., 2022*), lung cancer (*Villegas et al., 2002*), and gastric cancer (*Ishigami et al., 2000*). Recently, based on their anti-tumor potential, NK cells were exploited for treating malignancies and have proven to be highly promising for the treatment of certain hematologic malignancies (*Lamb et al., 2021*; *Liu et al., 2020*) but have limited efficacy for treating solid tumors (*Zhang et al., 2024*; *Hu et al., 2019*; *Oh et al., 2019*). Additionally, intratumor NK cells differ phenotypically or functionally from peripheral NK cells in several malignancies (*Bruno et al., 2013*; *Carrega et al., 2008*); a specific subset of tumor-infiltrating NK cells (CD11b[−] and CD27[−]) as well as decidual-like NK (CD9[+] and cd49a[+]) was associated with tumor progression in lung and hepatocellular cancers (*Zhang et al., 2017*; *Jin et al., 2013*; *Albini and Noonan, 2021*). Notably, the tumor-promoting transformation of NK cells may be educated by cancer cells (*Chan et al., 2020*; *Huergo-Zapico et al., 2018*). However, their function remains elusive. In particular, the presence of specific NK cell subsets within colon cancer and their potential relationship to CCLM progression have not been fully characterized to date.

In this study, we combined previously published single-cell RNA-sequencing, spatial transcriptomics (ST), and bulk RNA-sequencing data from public datasets, aiming to comprehensively chart the cellular landscape of TME in primary colon cancer and matched liver metastasis. Results of bioinformatics analysis were further validated through an in vitro co-culture experiment (*Figure 1A*).

## Results

### Integrated single-cell and spatial transcriptome RNA-seq precisely quantified immune cell diversity in CCLM

We used a previously published single-cell dataset containing 89 samples from paired samples of colon cancer, adjacent colon, liver metastasis, adjacent liver, lymph nodes along colons, and peripheral blood mononuclear cells from 20 patients to define the heterogeneous immune microenvironment landscapes of CCLM. Subsequently, 178,630 CD45+ cells were integrated.

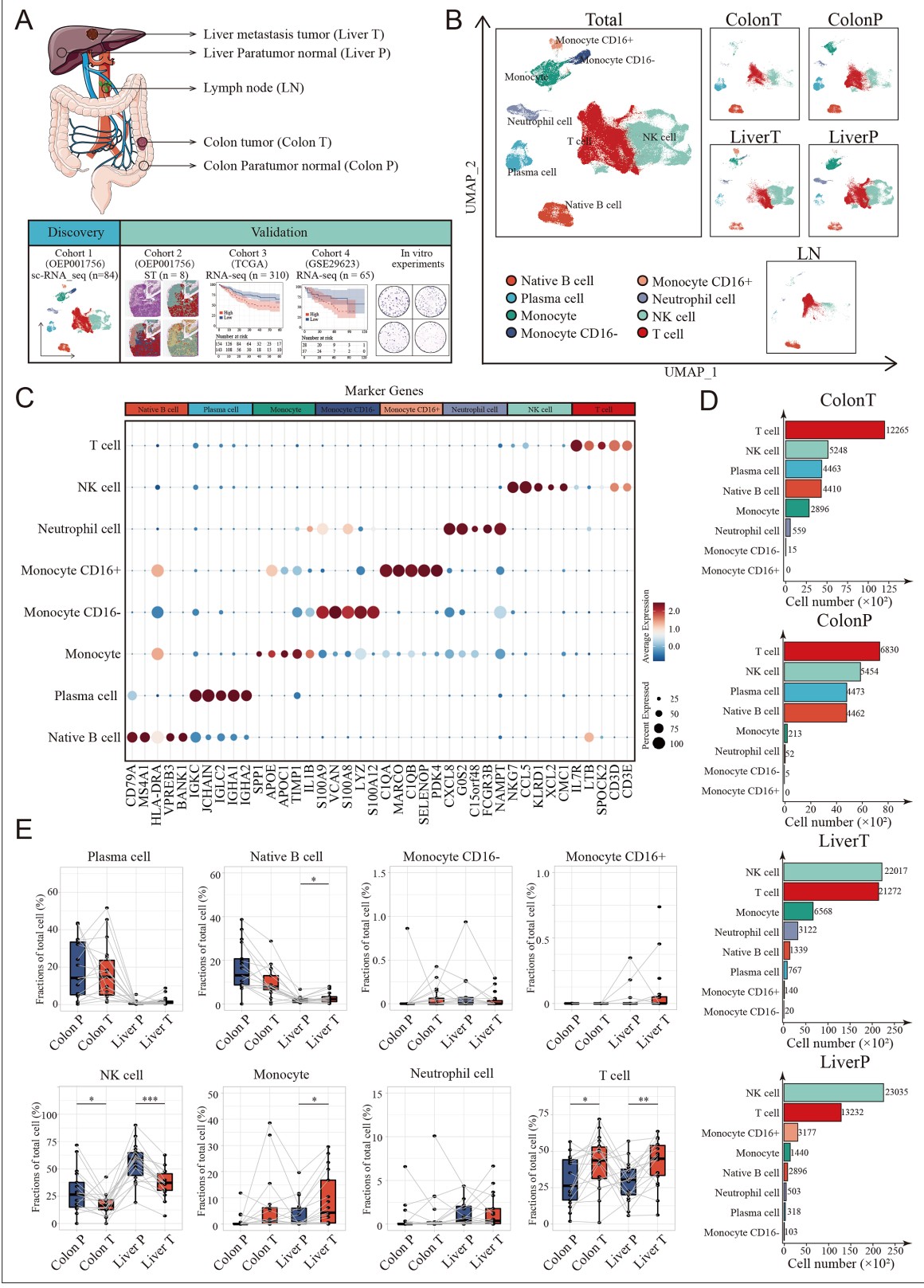

**Figure 1.** Landscape of tumor immune microenvironment in colon cancer liver metastasis (CCLM) revealed by single-cell transcriptomics. (**A**) Schematic overview of the experimental design and analytical workflow. Colon T: colon tumor tissue; Colon P: colon paratumor normal tissue; Liver T: liver metastasis tumor tissue; Liver P: liver paratumor normal tissue; LN: lymph node metastasis tissue. (**B**) The Uniform Manifold Approximation and Projection (UMAP) plot of all main immune cell types. (**C**) Dot plots showing average expression of highly variable genes of each cell group. The dot size

*Figure 1 continued on next page*

*Figure 1 continued*

represents percent of cells expressing the genes in each cluster and the color of dot represents the expression intensity. (**D**) The cell numbers of main immune cells across tissues. (**E**) Proportions of all main immune cells. p values were determined by the paired nonparameter test. (*: p<0.05, **:p<0.01, *** : p<0.001).

The online version of this article includes the following figure supplement(s) for figure 1:

**Figure supplement 1.** Cluster characterization of the global landscape of colon cancer liver metastasis (CCLM) revealed by single-cell transcriptomics.

To further define the main cell type, clustering analysis and SingleR were performed, and monocytes, neutrophil cells, native B cells, plasma cells, T cells, and NK cells were identified from the CCLM samples (*Figure 1B, C* and *Figure 1—figure supplement 1A, B*; *Supplementary file 1*). Notably, T and NK cells were the major components of all immune cells and distinct across different tissue types, with NK cells significantly decreasing in tumor tissues (both primary colon cancer and liver metastasis cancer) compared to corresponding adjacent normal tissues, whereas T cells were the opposite (*Figure 1D, E*; *Figure 1—figure supplement 1C*).

To comprehensively analyze the spatial distribution profile of colon primary tumors and CCLM tumors, we collected spatial transcriptome (ST) data, which came from eight samples including paired primary colon cancer and liver metastasis. Unsupervised clustering analysis was first performed to cluster similar ST spots, and the annotation of the clusters was further determined according to cell marker genes (*Figure 2—figure supplement 1A–C*, *Supplementary file 2*). Furthermore, after combining the gene expression features of each sample (*Figure 2—figure supplement 2*) and hematoxylin and eosin (H&E) staining, six morphological regions, including tumor, fibroblast, smooth muscle, B cells, hepatocytes, normal epithelium, and tumor and paratumor areas, were identified (*Figure 2A–C*).

To integrate the scRNA-seq and ST data, we used the AddModuleScore function in Seurat to quantify the main immune subpopulations. Consistent with scRNA-seq, T cells were remarkably enriched in the tumor region of both primary colon cancer and liver metastasis cancer, whereas NK cells were enriched in the non-tumor area (*Figure 2D*).

## Biological relationship between NK cells and metastasis of colon cancer revealed by bulk RNA transcriptomics

TME cell composition differences between metastasis and non-metastasis colon cancer were evaluated using multiple tools that could robustly quantify the abundance of cell populations based on transcriptomic datasets, including xCell, EPIC, quanTIseq, and MCPcounter and bulk-seq datasets (TCGA COAD cohort). The percentages of NK cells markedly decreased in metastasis colon cancer using MCPcounter, EPIC, and xCell (*Figure 3A–D*).

To further understand the potential triggers that induce colon cancer metastasis, the differentially expressed genes (DEGs) were calculated between the metastasis and non-metastasis colon cancer groups in the TCGA COAD cohort. A total of 1378 DEGs were determined using the limma package with cutoffs |log fold change| > 1.5 and p < 0.05, including 817 upregulated and 561 downregulated genes between metastasis and non-metastasis colon cancer. Subsequently, Gene Ontology (GO) and Kyoto Encyclopedia of Genes and Genomes (KEGG) enrichment analyses were performed to investigate the correlative functions and pathways, As shown in *Figure 3E*, cell killing, NK cell-mediated immunity, NK cell-mediated cytotoxicity, NK cell activation, and MHC-related GO terms and the KEGG pathways of NK cell-mediated cytotoxicity were significantly enriched. Interestingly, nearly all the genes enriched in pathway of NK cell-mediated cytotoxicity were markedly downregulated in the metastasis group compared to the non-metastasis group (*Figure 3F*). Similarly, gene set enrichment analysis (GSEA) revealed that NK cell-mediated cytotoxicity was enriched in the non-metastasis group (*Figure 3G, H*).

## The landscape of NK cells in CCLM progression

Considering the differences in infiltrated NK cells between tumor-adjacent tissues and tumor tissues as well as colon cancer with and without metastasis, we further focused on NK cells to explore the detailed difference of NK subgroups in the CCLM microenvironments. Unsupervised clustering analysis of NK cells was thus performed and eight NK cell subtypes were thus identified (*Figure 4A*,

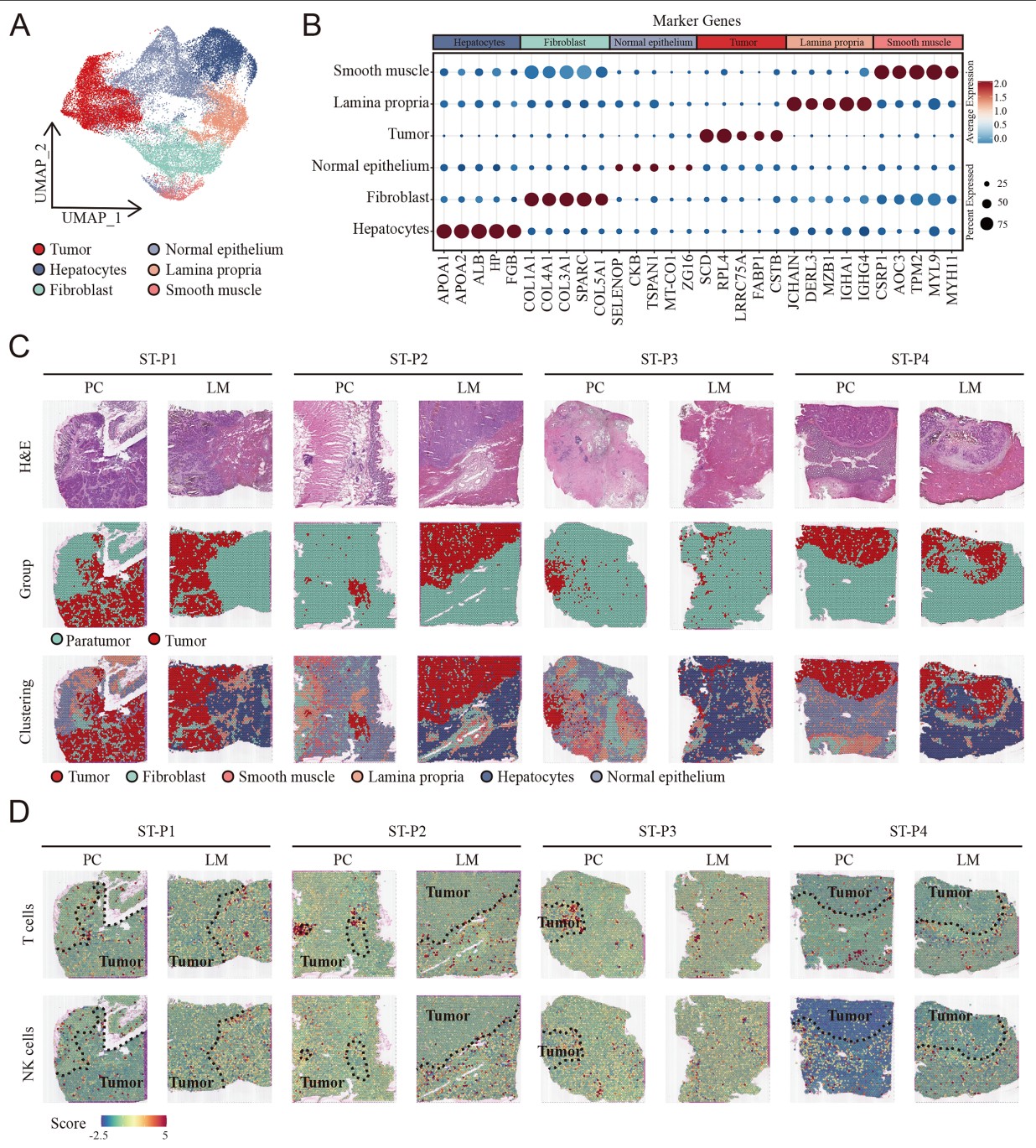

**Figure 2.** Cellular identification in spatial transcriptomic samples. (**A**) UMAP visualization of cell clusters in spatial transcriptomic samples. (**B**) Dot plots showing average expression of markers in indicated cell clusters. (**C**) Overview of the spatial transcriptomic sections. Hematoxylin and eosin (H&E) staining of spatial transcriptomic sections (upper). Tumor tissue and paratumor tissue identification of each section (middle). Spatial cluster distribution of each section (lower). PC: primary cancer; LM: liver metastasis; ST-P1: spatial transcriptomic dataset patient 1; ST-P1: spatial transcriptomic dataset patient 1; ST-P2: spatial transcriptomic dataset patient 2; ST-P3: spatial transcriptomic dataset patient 3; ST-P4: spatial transcriptomic dataset patient 4. (**D**) The signature scores of T cells (upper) and natural killer (NK) cells (lower) in colon cancer and liver metastasis in the spatial transcriptomic sections.

The online version of this article includes the following figure supplement(s) for figure 2:

**Figure supplement 1.** Cellular identification in spatial transcriptomic samples.

**Figure supplement 2.** Gene expression features of each sample.

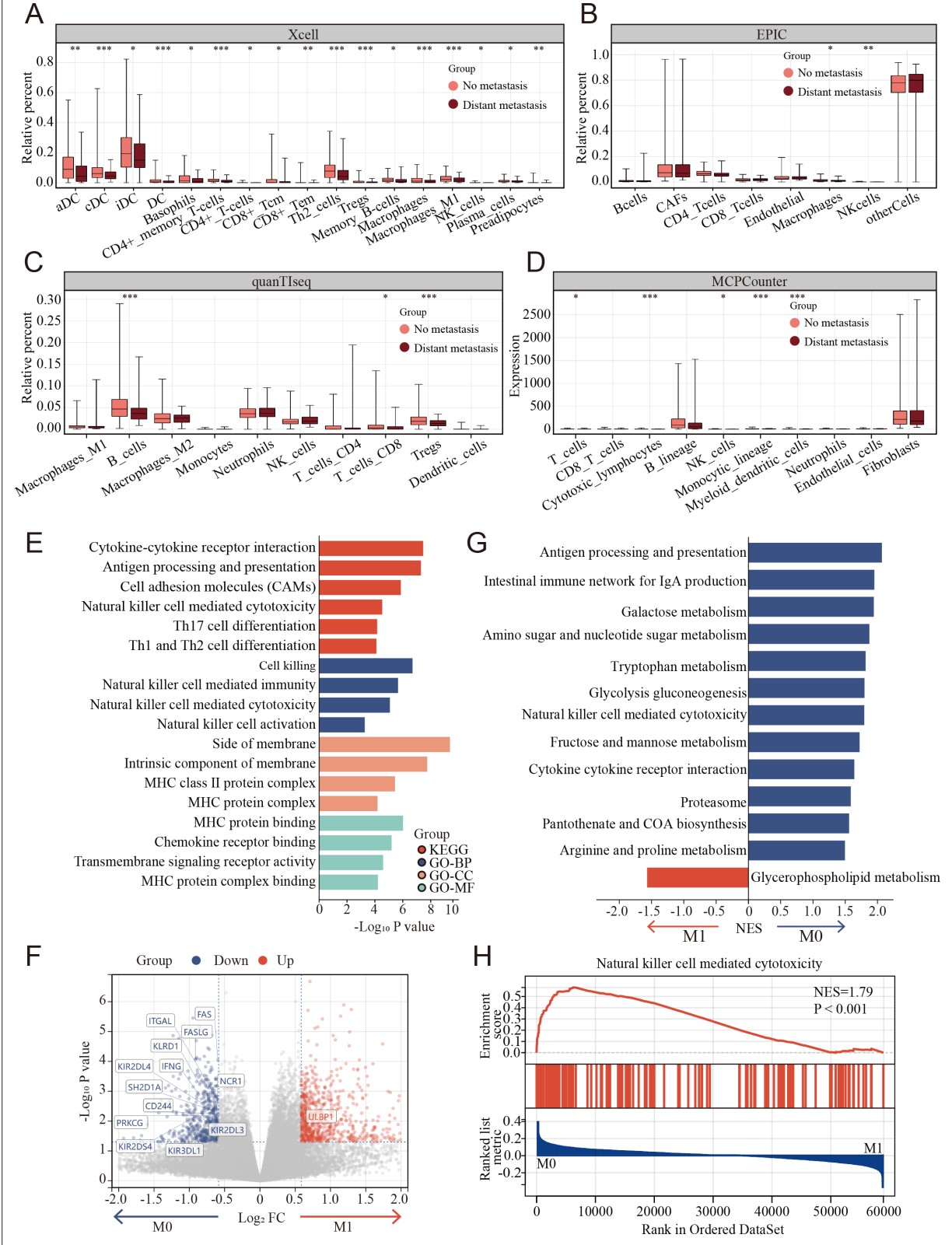

**Figure 3.** Clinical and biological relationship between natural killer (NK) cells and metastasis of colon cancer revealed by bulk RNA transcriptomics. (A–D) The relationship of immune cell percentage determined by xCell, EPIC, quanTIseq, and MCPCounter between metastasis and non-metastasis tumor in TCGA COAD cohort. (*: p<0.05, **:p<0.01, *** : p<0.001) (E) Gene Ontology (GO) and Kyoto Encyclopedia of Genes and Genomes (KEGG) enriched pathway bar chart of differentially expressed genes (DEGs) in metastasis versus non-metastasis colon cancer. (F) Volcano plot showing nearly

*Figure 3 continued on next page*

Figure 3 continued

all the genes enriched in pathway of NK cell-mediated cytotoxicity were downregulated in the metastasis colon cancer. (**G**) Gene set enrichment analysis (GSEA) of KEGG gene set. (**H**) Natural killer cell-mediated cytotoxicity was enriched in the non-metastasis colon cancer.

*Supplementary file 3*). Combining the highly variable features of each NK cluster (*Figure 4B*) and CIBERSORT-reported canonical NK cell markers (*Figure 4C, D*), NK cells were classified into three cell types, including activated NK cells identified by the expression of KIR2DL4, GPR183, GRP171, CD69, and IFNG, resting NK cells marked by GZMK, TTC38, CD160, and PLEKNF1, and the other NK cells of which no characteristic gene was identified. The marker genes of three NK cell subtypes were also identified (*Supplementary file 4*). Although all three NK cell subtypes were presented in primary colon tumors, adjacent normal colon tissues, liver metastasis tumors, and lymph nodes (*Figure 4E*), the infiltration grade for each of these NK cell subsets was significantly different. At the individual sample level, considerable variability was observed in the NK cell subset composition. Where serial samples were available from individual patients, the expression of resting NK cells increased, whereas that of activated NK cells decreased during disease progression (*Figure 4F, G*). Additionally, using the paired primary colon tumor, adjacent normal colon tissues, and liver metastasis tumor samples, we observed that the gradient of the resting NK cells increased, whereas the gradient of the activated NK cells decreased (*Figure 4H*).

To evaluate the clinical relationship between NK cell differences and CCLM, CIBERSORT was performed using the TCGA COAD cohort. The percentages of resting NK cells significantly increased (*Figure 5A*), and the activated NK cells decreased in the metastasis group (*Figure 5B*), which is consistent with the scRNA-seq results. Also, we observed that the resting NK cells were significantly increased, whereas the activated NK cells decreased in higher T, N, and TNM stages (*Figure 5—figure supplement 1A*). In terms of survival, neutrophils and resting NK cells were associated with a worse prognosis, whereas activated NK cells and M1 macrophages were associated with better outcomes (*Figure 5—figure supplement 1B*). Furthermore, survival analysis showed that colon cancer patients with lower and higher degrees of infiltration of activated and resting NK cells, respectively, had a significantly worse prognosis in the TCGA COAD and GSE29623 cohorts (*Figure 5C, D*). Interestingly, resting NK cells were remarkably enriched in the tumor region, whereas activated NK cells were enriched in the non-tumor area of both primary colon cancer and liver metastasis (*Figure 5E*).

## Characterization and developmental course of differential subsets of NK cells in CCLM

To further study the heterogeneity of NK cells subgroup, the proportions of all eight clusters of NK cells were calculated (*Figure 6—figure supplement 1*). KIR2DL4+ activated NK cells were lessened in proportion with the progression of the disease from normal colon to colon cancer and liver metastasis cancer, whereas GZMK+ resting NK cells increased (*Figure 6A, B*), suggesting that these two NK subsets were associated with metastasis. ST analysis also confirmed that there was more infiltration of the KIR2DL4+ activated NK cell subset in the non-tumor area and GZMK+ resting NK cells in the tumor region, both primary colon cancer and liver metastasis (*Figure 6C*). We next utilized mIHC to check those NK cells in the clinical tissue samples of a colon cancer patient with liver metastasis, confirming a high infiltration of GZMK+ NK cells and a low infiltration of KIR2DL4+ NK cells in live metastasis cancer (*Figure 6D*).

Pseudotime cell trajectory analysis of the two NK cell clusters was constructed to investigate the evolutionary dynamics of metastasis-associated NK cells. NK cells from the normal colon group were located at the top right corner of the trajectory curve, suggesting the clear starting point of this evolving trajectory curve. After confirmation of this starting point, developmental routes were determined to begin with the KIR2DL4+ activated NK cells and then develop into GZMK+ resting NK cells (*Figure 6E*).

To identify the cell cluster marker genes that change as NK cell subtypes move from the activated stage to the resting stage, Branch expression analysis modeling (BEAM) analysis was performed. The branched heatmap showed the expression dynamics of the top 200 significant genes in different cell fate branches. Corresponding GO enrichment analyses of these significant genes further demonstrated that a mass of leukocyte differentiation and immune system process-related GO terms were significantly enriched (*Figure 6F*).

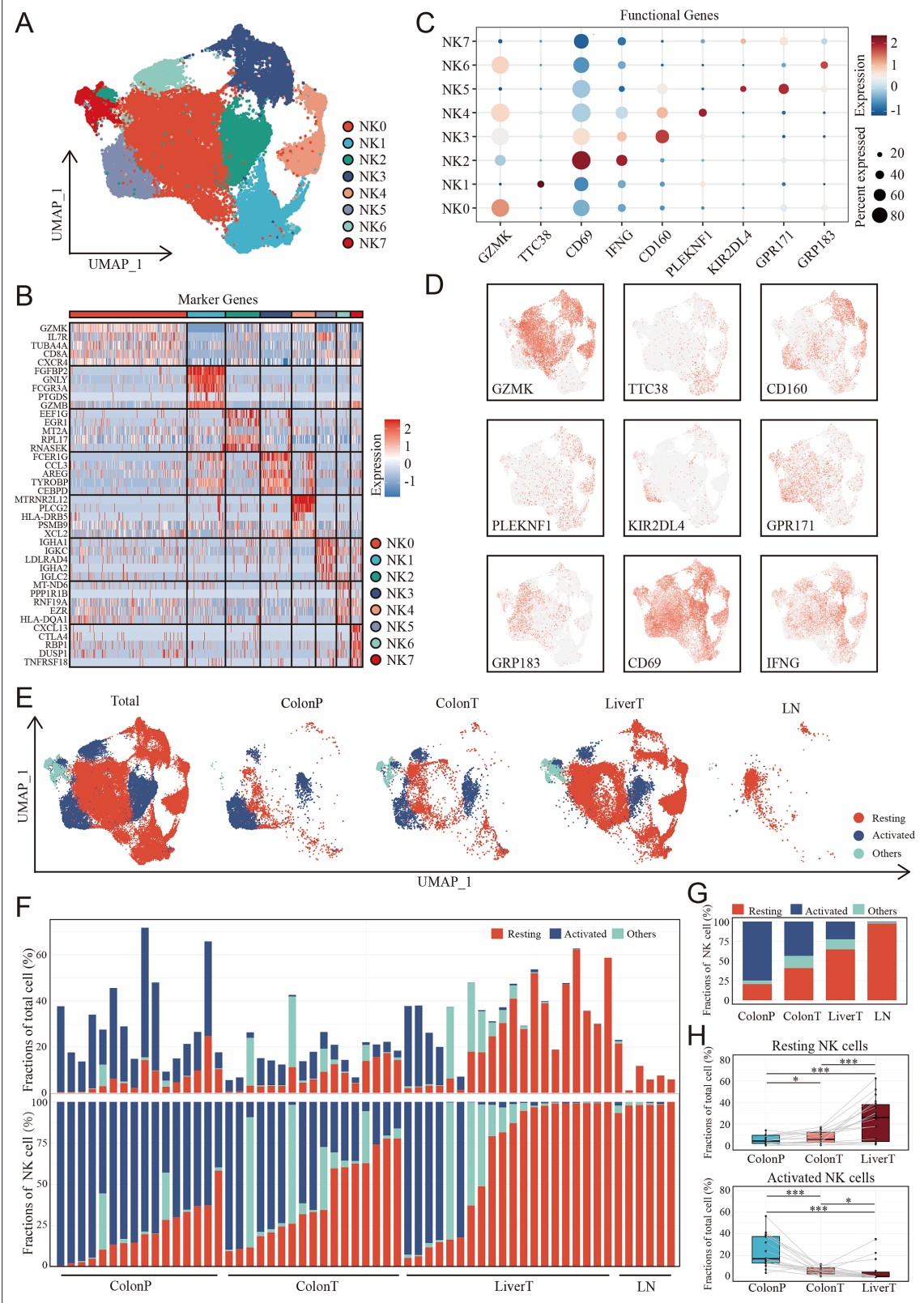

**Figure 4.** The landscape of natural killer (NK) cells in the disease progression of colon cancer liver metastasis (CCLM). (**A**) The UMAP plot of NK cells from CCLM. (**B**) Unsupervised clustering identifies eight subsets of NK cells. (**C**) Expression of key markers that distinguish resting and activated subsets of NK cells. (**D**) Expression of key resting and activated NK cell markers across all samples. (**E**) The UMAP plot of distribution of resting and activated NK cells from ColonP, ColonT, LiverT, and LN. (**F**) Cellular landscape of each sample from the ColonP, ColonT, LiverT, and LN. The proportion of NK

*Figure 4 continued on next page*

*Figure 4 continued*

cells subsets in total immune cells (upper). The proportion of NK cells subsets in total NK cells (lower). (**G**) Number of cells identified from each group (ColonP, ColonT, LiverT, and LN) by cell type proportion. (**H**) Proportions of resting (upper) and activated (lower) subsets of NK cells. p values were determined by the paired nonparameter test. (*: p<0.05, **:p<0.01, *** : p<0.001).

Interestingly, patients with colon cancer with higher infiltration of GZMK+ resting NK cells (*Figure 6G*) and lower infiltration of KIR2DL4+ activated NK cells (*Figure 6H*) exhibited shorter survival in the TCGA COAD cohort.

## Tumor cell-educated NK cells shift toward tumor-promoting status depends on cell-to-cell interaction

To verify the NK cell-mediated inductive effect on colon cells, we set up mixed cell co-culture experiments using colon cell line HCT-116 and NK cell line NK-92 in the ratio 1:1 (CN group). As controls, NK cells were cultured alone, either in the tumor supernatant (SN group) or fresh medium (MN group). The co-cultured supernatant of each group was collected to incubate fresh HCT-116 after 24 hr of co-culturing. Interestingly, Cell Counting Kit-8 (CCK-8) and colony formation assays showed that colon cells in CNS group, which colon cells were cultured in the supernatant of co-cultured system that NK and cancer cell were in contact, underwent a significant increase in proliferation (*Figure 7A–C*) compared to the CS group, where cancer cells were cultured directly in the supernatant of cancer cell after 24 hr culturing. Similarly, the migration and invasion of HCT-116 were also significantly increased in CNS group (*Figure 7D–F*). Furthermore, colony formation assays (*Figure 7G, H*), CCK-8 (*Figure 7I*), and transwell assays (*Figure 7J–L*) showed colon cells in CNS group underwent a significant increase in proliferation, invasion and migration, compared to those cultured in the co-cultured supernatant (SNS group) and fresh medium (MNS group).

Additionally, another colon cancer cell line, DLD-1 was chosen to evaluate the effect of the supernatant in the different co-culture groups. DLD-1 cells in the CNS group did not undergo a prominent increase in proliferation compared to CS group (*Figure 7—figure supplement 1A–C*). The same results were obtained for their migration and invasion (*Figure 7—figure supplement 1D–F*). Similarly,

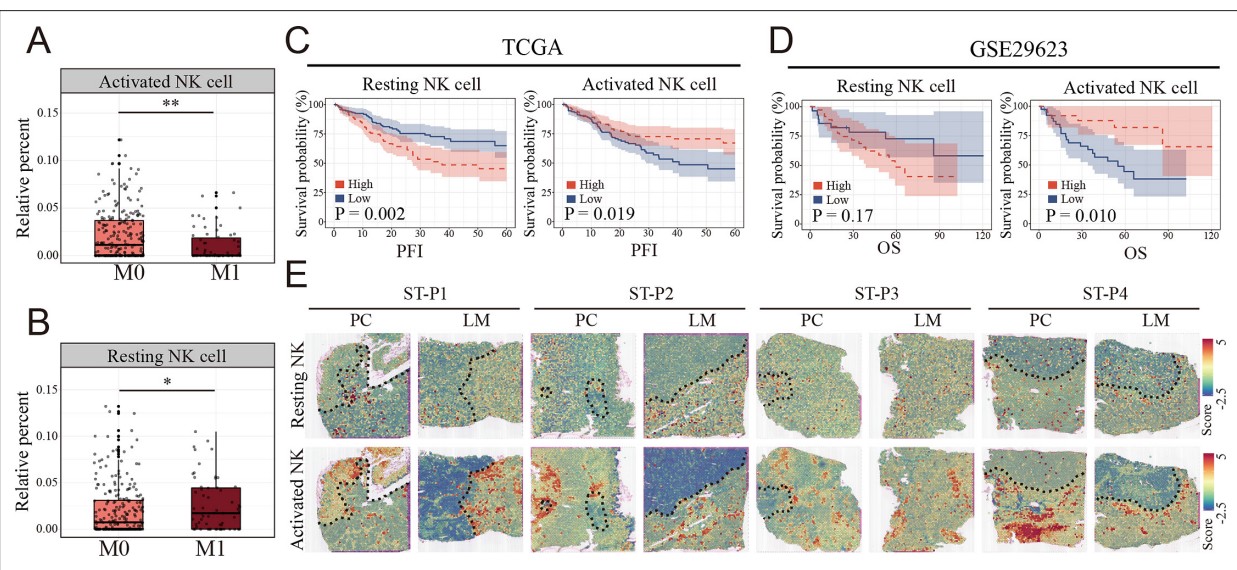

**Figure 5.** Clinical relationship between natural killer (NK) cells subsets and metastasis of colon cancer revealed by bulk RNA and spatial transcriptomics. (**A, B**) The relationship of activated and resting NK cell percentage determined by CIBERSORT and tumor metastasis in TCGA COAD cohort. M0: non-metastasis colon cancer; M1: metastasis colon cancer. (*: p<0.05, **:p<0.01, *** : p<0.001) (**C, D**) K–M survival plots show that high resting NK cell and low activated NK cell predicted poor prognosis in TCGA COAD and GSE29623 cohort. PFI: progression-free interval; OS: overall survival. (**E**) The signature scores of resting (upper) and activated (lower) NK cells in colon cancer and liver metastasis in the spatial transcriptomic sections.

The online version of this article includes the following figure supplement(s) for figure 5:

**Figure supplement 1.** Clinical relationship between natural killer (NK) cells subsets and colon cancer revealed by TCGA COAD cohort.

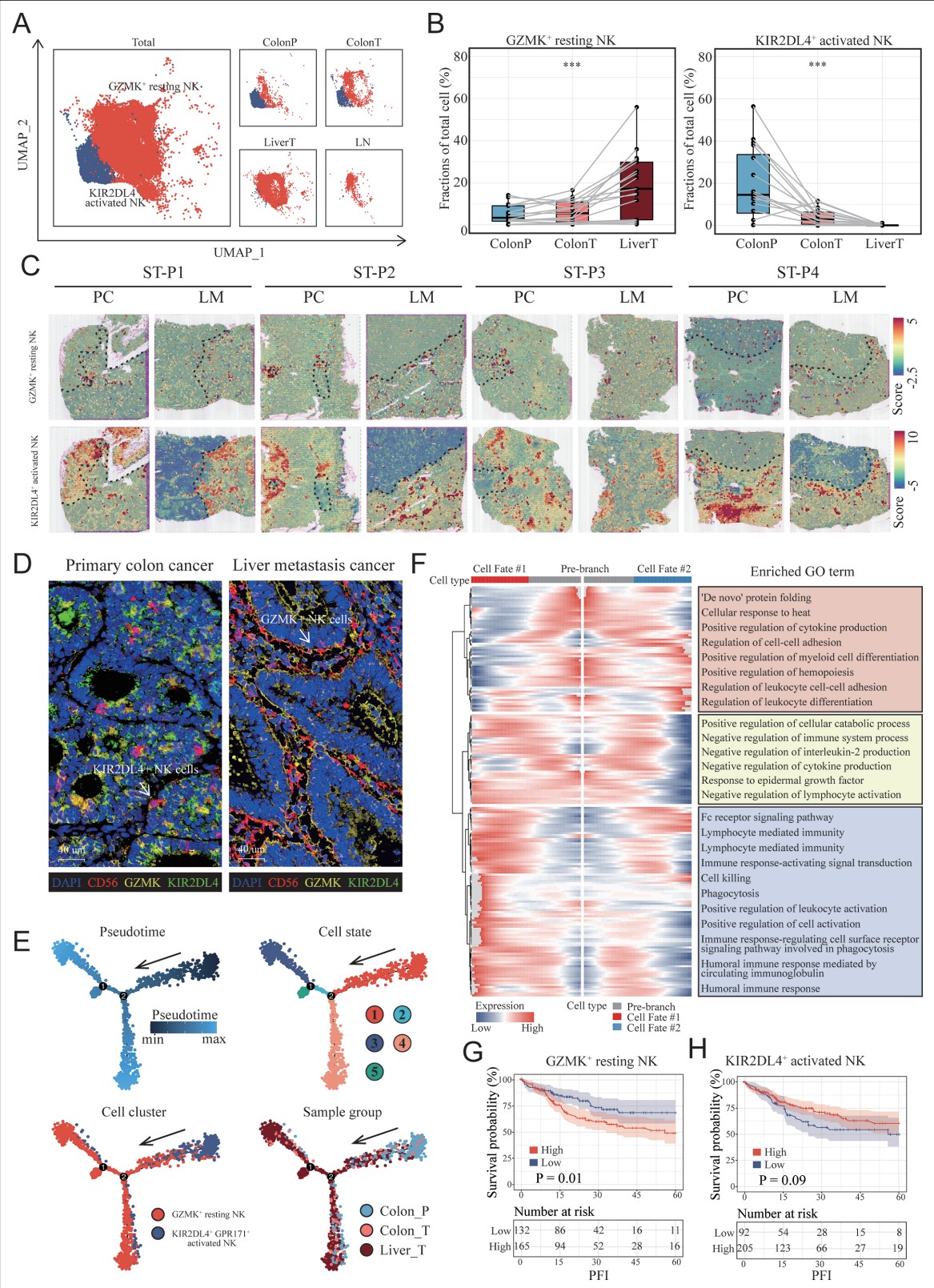

**Figure 6.** Characterization and developmental course of differential subsets of natural killer (NK) cells in colon cancer liver metastasis (CCLM). (**A**) The UMAP plot of KIR2DL4+ activated NK cells, GZMK+ resting NK cells from CCLM. (**B**) Proportions of KIR2DL4+ activated NK cells, GZMK+ resting NK cells in total immune cells. p values were determined by the paired nonparameter test. (*: p<0.05, **:p<0.01, *** : p<0.001) (**C**) The signature scores of GZMK+ resting NK cells (upper) and KIR2DL4+ activated NK cells (lower) in colon cancer and liver metastasis in the spatial transcriptomic sections.

*Figure 6 continued on next page*

*Figure 6 continued*

(**D**) mIHC of GZMK+ and KIR2DL4+ NK cells in primary colorectal cancer and liver metastasis cancer. (**E**) Monocle analysis showing the developmental trajectory of NK cells. Color as in pseudotime, cell state, subsets of NK cells, and sample group. (**F**) The heatmap shows the expression patterns of the top 50 significant genes in branched expression analysis modeling, associated Gene Ontology (GO) terms (using DAVID v6.7) are given on the right of the corresponding gene clusters. (**G**, **H**) The Kaplan–Meier curve shows COAD patients survival with different GZMK+ resting NK cells and KIR2DL4+ activated NK cells infiltration.

The online version of this article includes the following figure supplement(s) for figure 6:

**Figure supplement 1.** The proportions of eight clusters of NK cells in Colon P, Colon T, Liver T and LN.(*: p<0.05, **:p<0.01, *** : p<0.001).

no such cell functional changes were induced in the CNS group compared to SNS and MNS groups (*Figure 7—figure supplement 1G–L*).

Given the characteristic spatial distribution of NK cell subpopulations, we hypothesized that the tumor region-enriched NK cells might be educated by tumor cells and thus functionally distinct. To verify the possible effect of tumor cells on the phenotypic switch of NK cells, NK cells were analyzed using FACS to determine the expression of the NK cells markers above identified (GZMK and KIR2DL4) as well as tumor-infiltrated NK cells markers: CD9, CD49a, and PD-1. Fluorescence Minus One (FMO) control was set up to define the position of the positive gate as shown in *Figure 7—figure supplement 2*. Upon exposure to tumor cells, NK cells significantly decreased and slightly increased the expression of activated marker KIR2DL4 (*Figure 7M*) and the resting marker GZMK (*Figure 7N*), respectively. On the other way, for GZMK, which known as the secretory protein, flow cytometric analyses were also performed with cell fixation and permeabilization. However, nearly all the NK cells in different groups were GZMK positive and no significant differences were found among each group (*Figure 7—figure supplement 3*). Additionally, NK cells in CN group underwent an increasing of tumor-infiltrated NK cells markers, CD9 (*Figure 7O*), CD49a (*Figure 7P*), and PD-1 (*Figure 7Q*), especially CD9. However, no such effect was observed after co-culturing with tumor supernatant and fresh medium.

On the other way, NK cells were also co-cultured with HFF-1, a normal human cell line. The tissue infiltrated NK phenotype identified above (CD9, CD49a, and PD-1) were determined. When co-cultured with HFF-1 in direct contact (CN group), NK cells were also tending toward tissue infiltration state. However, the domestication effect was significantly reduced compared to co-culturing with tumor cells (*Figure 7—figure supplement 4A, B*). Additionally, fresh HCT underwent a limited increase (no statistical significance was found) in migration when cultured in CNS group, but not in the SNS group and MNS group (*Figure 7—figure supplement 4C-E*).

## Resting NK-induced colon cancer malignant phenotype promotion depends on SCF release

To study the potential mechanism by which the supernatant from the NK and colon cells in direct contact with the co-culture system promoted colon cell migration, Luminex liquid suspension chip detection was used to compare the differential expression of 48 common chemotactic and inflammatory cytokines in the CNS, SNS, and MNS groups. We observed that SCF was upregulated in all three repetitions in the CNS group (*Figure 8A, B*). Furthermore, enzyme-linked immunosorbent assay (ELISA) confirmed the upregulated level of SCF in the CNS group (*Figure 8C*).

We further assessed the inhibitory effects of imatinib mesylate on the enhanced proliferation and invasion. CCK-8 (*Figure 8D, E*) and colony formation assays (*Figure 8F–I*) showed that imatinib mesylate (2 μM) significantly inhibited supernatant-enhanced proliferation of HCT-116. Similarly, the supernatant-enhanced migration (*Figure 8J, K*) and invasion (*Figure 8L, M*) were also inhibited by imatinib.

## Discussion

CCLM is a multistep process and is mostly fatal (*Manfredi et al., 2006*). During disease progression, functional interactions between tumor cells and the surrounding TME are critical for influencing tumor growth, promoting angiogenesis, and finally resulting in metastasis (*Quail and Joyce, 2013*). Since the genomic divergence between primary and metastatic CRC cells is relatively low (*Wei et al., 2017*; *Yaeger et al., 2018*) and accumulating evidence has highlighted the key role of tumor-infiltrating immune cells in dictating the fate of cancer cells (*Kumar et al., 2018*; *Hanahan and Coussens,*

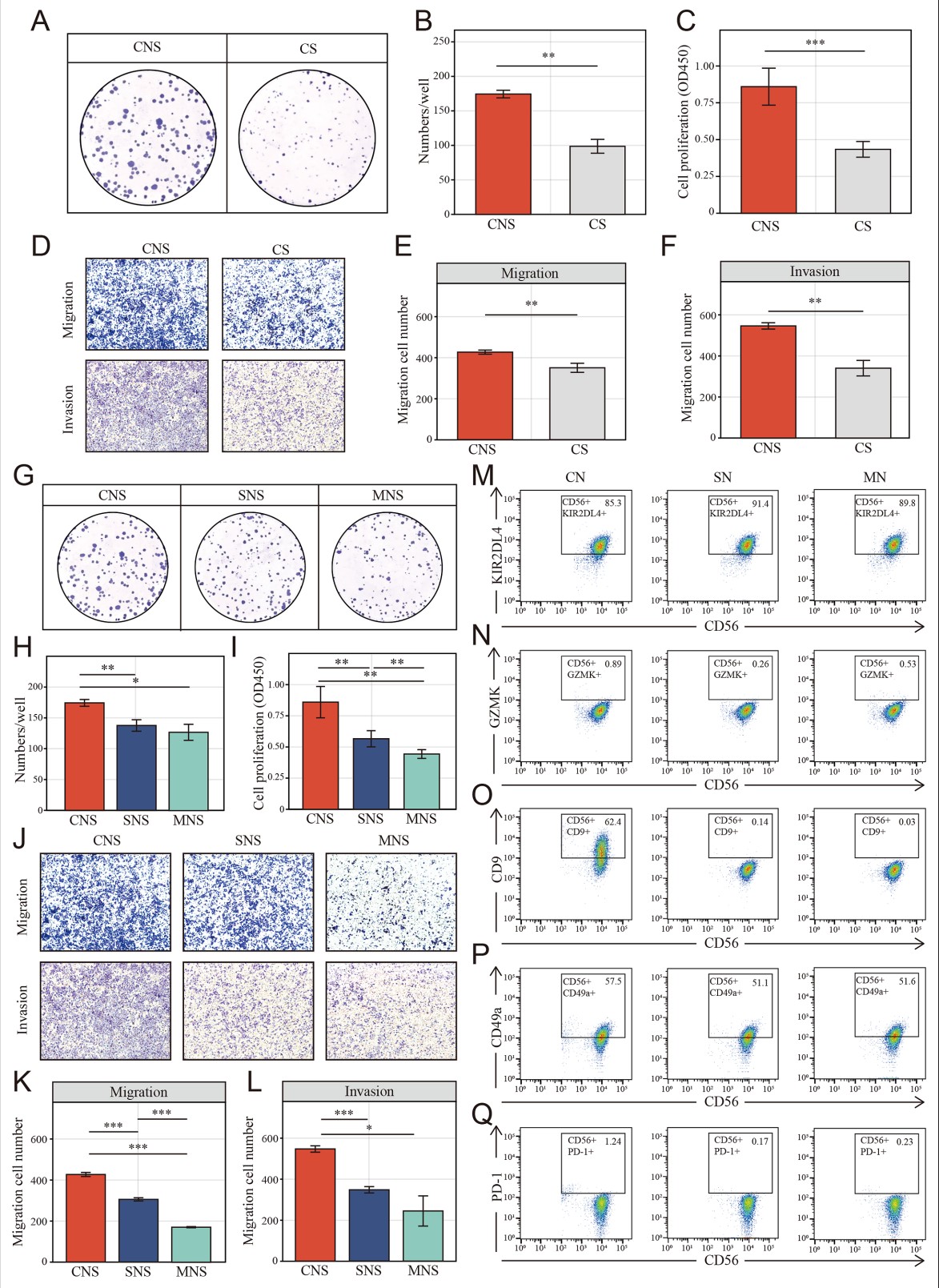

**Figure 7.** Colon cancer cells (HCT-116) educated NK cells shift toward tumor-promoting status depends on cell-to-cell interaction. (**A, B**) Clone formation assay showed the natural killer (NK) cell-mediated inductive effect on cell proliferation of colon cancer cell (HCT-116). Colon cancer cells were cultured in the supernatant from different co-culture system in transwell devices. CNS: colon cancer cells were cultured in the supernatant from co-culture system that NK cells and cancer cells were cultured in direct contact (CNS); CS: cancer cells cultured directly in supernatant of cancer cells.

*Figure 7 continued on next page*

*Figure 7 continued*

(**C**) Cell Counting Kit-8 (CCK-8) assay showed the NK cell-mediated inductive effect on cell proliferation of colon cancer cell among CNS and CS groups. (**D–F**) The NK cell-mediated inductive effect on migration and invasion of colon cancer cell among CNS and CS groups. (**G-I**) Clone formation assay and CCK-8 assay showed the NK cell-mediated inductive effect on cell proliferation of colon cancer cell among CNS, SNS, and MNS groups. SNS: colon cancer cells were cultured in the supernatant from co-culture system that NK cells were cultured in supernatant of cancer cells; MNS: colon cancer cells were cultured in the supernatant from co-culture system that NK cells were cultured in fresh medium. (**J-L**) The NK cell–mediated inductive effect on on migration and invasion of colon cancer cell (HCT-116) among CNS, SNS, and MNS groups. (**M-Q**) Phenotype switch (KIR2DL4, GZMK, CD9, CD49a, and PD-1) of NK cells was induced by cell-to-cell interactions with cancer cells. CN: NK cells were co-cultured with colon cancer cells; SN: NK cells were cultured in supernatant of cancer cells; MN: NK cells were cultured in fresh medium. (*: p<0.05, **:p<0.01, *** : p<0.001).

The online version of this article includes the following figure supplement(s) for figure 7:

**Figure supplement 1.** Natural killer (NK) cell-mediated tumor-promoting effect in colon cancer cells (DLD-1).

**Figure supplement 2.** The positive gate locations of CD56, GZMK, KIR2DL4, CD9, CD49a, and PD-1 defined according to the Fluorescence Minus One (FMO) control.

**Figure supplement 3.** Phenotype switch (CD56+, GZMK+) of natural killer (NK) cells was analyzed by fluorescence-activated cell sorting (FACS) after fixation and permeabilization in different co-cultured groups.

**Figure supplement 4.** Phenotype switch of natural killer (NK) cells in different co-cultured system and the corresponding NK cell-mediated effect on cell migration of fresh colon cancer cell (HCT-116).

**Figure supplement 5.** Schematic overview of the in vitro experimental design.

*2012*), we presented a comprehensive cellular and spatial immune landscape of the primary and liver metastatic tumors of CRC by using previously published scRNA-seq and ST data. Consistent with previous studies (*Zhou et al., 2021*; *Wu et al., 2022*), we observed that the immune microenvironment of primary and liver metastasis lesions of colon cancer had undergone extensive remodeling with a significantly decreased proportion of NK cells and a strong enrichment of T cells. However, the contrast between the decreased proportion of NK cells in tumor tissue and the enrichment of NK cells in tumor regions may indicate the specific effects of NK cells in CCLM.

Traditionally, NK cells within the TME were known as potential suppressors for tumor growth by directly killing cells and secreting proinflammatory cytokines (*Huntington et al., 2007*; *Freud and Caligiuri, 2006*). However, the function of NK cells highly depends on their maturation status and localization (*Cooper et al., 2001*). The peripheral blood CD56$^{dim}$CD16$^{high}$ NK cell population predominantly mediates the killing of target cells by secreting perforin and granzymes (*Jacobs et al., 2001*), whereas the CD56$^{bright}$CD16$^{low}$ NK cells that reside in secondary lymphoid tissues are immature and have reduced cytotoxic capability (*Moretta, 2010*). Recently, tumor-associated NK cells, which are enriched in tumors with impaired anti-tumor functions, were discovered to be associated with an unfavorable prognosis and resistance to immunotherapy in multiple solid tumors (*Tang et al., 2023*), such as lung cancer, pancreatic cancer, and esophageal cancer. The potential impact of NK cells on CCML progression is still poorly investigated.

In this study, the proportion of NK cells was higher than in past cognition, similar to a previous study (*Xia et al., 2022*) in that NK cells occupied the largest immune cell compartment of nearly 50% in cholangiocarcinoma. In contrast, NK cells may not be well distinguished during automatic annotation using SingleR. Consistent with the study that revealed that treatment responders are associated with TME remodeling, including NK cell recruitment (*Kim et al., 2022*), the proportions of NK cells in PR patients were significantly higher among the LiverP and LiverT groups (*Figure 1—figure supplement 1D, E*), which also increased the overall proportion of NK cells to some extent. Additionally, we observed that NK cells are decreased in primary colon cancer and liver metastatic using scRNA-seq, consistent with previous findings supporting that NK cells play a crucial role in anticancer immunity (*Laskowski et al., 2022*; *Myers and Miller, 2021*). However, we further subdivided NK cells considering the conflicting result, in which NK cells were significantly enriched in the tumor area when using ST. Since this is the first study to comprehensively explore the heterogeneity of NK cells, no previous NK cell subpopulation markers were available; resting and activated NK cell markers in CIBERSORT were thus used to annotate NK cells. Consistent with a recent study that reported that mature and immature NK cells serve different functions (*Thacker et al., 2023*), our results also provide evidence that resting NK cells in the colon cancer TME possess a potential tumor-promoting activity, whereas activated NK cells play an anti-tumor role as conventional cognition. Bulk RNA-sequencing and the corresponding clinical data further indicated that higher infiltration of resting NK cells and

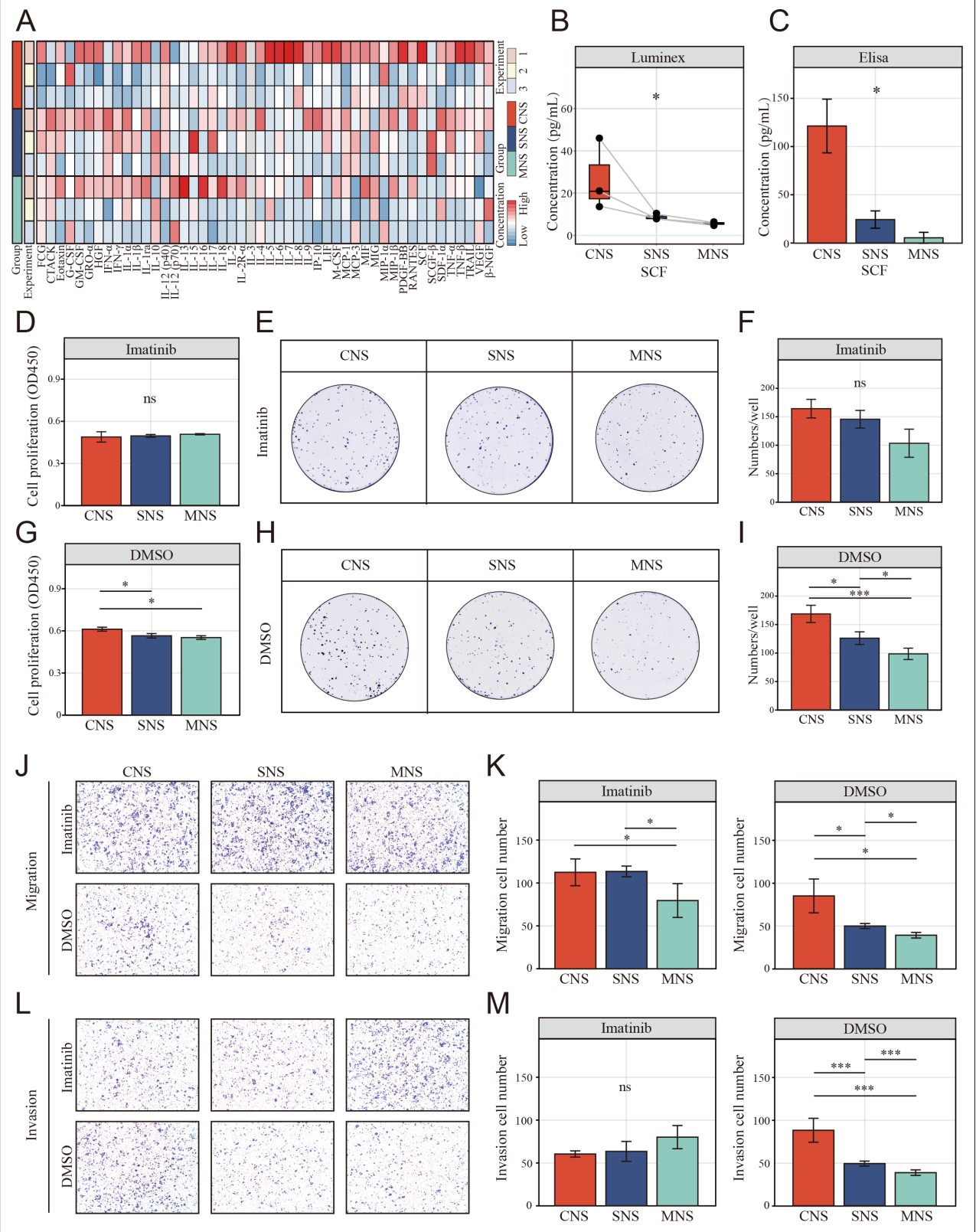

**Figure 8.** The resting natural killer (NK) cell promotes tumor malignant phenotype via elevating tumor-derived sSCF. (**A**) Luminex liquid suspension chip detection of 48 common chemotactic and inflammatory cytokines in CNS, SNS, and MNS group. (**B, C**) Concentration of SCF determined by luminex liquid suspension chip and ELISA in CNS, SNS, and MNS group. (**D, E**) Cell Counting Kit-8 (CCK-8) assay showed the proliferation of HCT-116 cells was inhibited by imatinib mesylate, evaluated by a CCK-8 assay. Cells were incubated in the supernatant from different co-culture system with DMSO or

*Figure 8 continued on next page*

*Figure 8 continued*

2 µM imatinib mesylate. (**F–I**) Clone formation assay showed the proliferation of HCT-116 cells was inhibited by imatinib mesylate. (**J–M**) The NK cell-mediated inductive effect on migration and invasion of HCT-116 was inhibited by imatinib mesylate. (*: p<0.05, **:p<0.01, *** : p<0.001).

The online version of this article includes the following figure supplement(s) for figure 8:

**Figure supplement 1.** Relative KITLG expression in different groups.

lower infiltration of activated NK cells were correlated with a worse prognosis in patients with CRC. However, since the resting and activated NK cells we identified were only the aggregation of multiple NK cell subpopulations, further studies are required to investigate the key NK cell subpopulations that play a crucial role in CCLM.

Further analysis of NK cell subtypes revealed a dramatic change in KIR2DL4+ activated NK cells and GZMK+ resting NK cells, whereas no other NK cell subpopulations differed in colon cancer progression. KIR2DL4, also known as CD158d, is an unusual member of the killer cell Ig-like receptor family expressed in all NK cells (*André et al., 2001*). Studies revealed that KIR2DL4 activates the cytotoxicity of NK cells (*Faure and Long, 2002*), and NK-92 cells stimulated with KIR2DL4 had higher cytotoxicity against breast cancer cells (*Kilic et al., 2023*). Consistent with previous studies, we also observed that the expression of the KIR2DL4+ activated NK cell subtype was decreased during CCLM progression, possibly reflecting the anti-tumor activity of this population. However, the GZMK+ resting NK cell subsets were first identified as potential tumor promoters. Moreover, it has been proven that GZMK[high] CD8[+] T effector memory cells, a cell subset that is particularly similar to GZMK+ NK cells, were associated with poor clinical outcomes in patients with colorectal tumors. Our results also showed that GZMK+ resting NK cells highly infiltrate the cancer region, and this predicts a worse prognosis in patients with colon cancer. However, despite the interesting and gratifying clinical relevance, the mechanism remains unknown.

Considering the evolutionary trajectory and the characteristic distribution of these metastasis-associated NK cell subtypes, we predicted that activated NK cells might differentiate into resting NK cells under the action of tumor cells. Consistent with a recent study that indicated that exposure to cancer cells causes NK cells to lose their cytotoxic ability (*Chan et al., 2020*), we identified the tendency of NK cells to differentiate to an resting state with decreasing expression of activated marker KIR2DL4 as well as increasing expression of decidual-like NK cells markers: CD9, CD49a, and PD-1 (*Albini and Noonan, 2021*), which are only triggered by cell-to-cell interactions as validated by FACS. It is worth mentioning that decidual-like NK cells were the newly identified subset of NK cells that promote tumors (*Albini and Noonan, 2021*).

Additionally, NK-mediated tumor cell editing appears to partly depend on the release of cytokines in in vitro co-culture experiments. SCF, the natural ligand of c-Kit with membrane-bound (mSCF) and

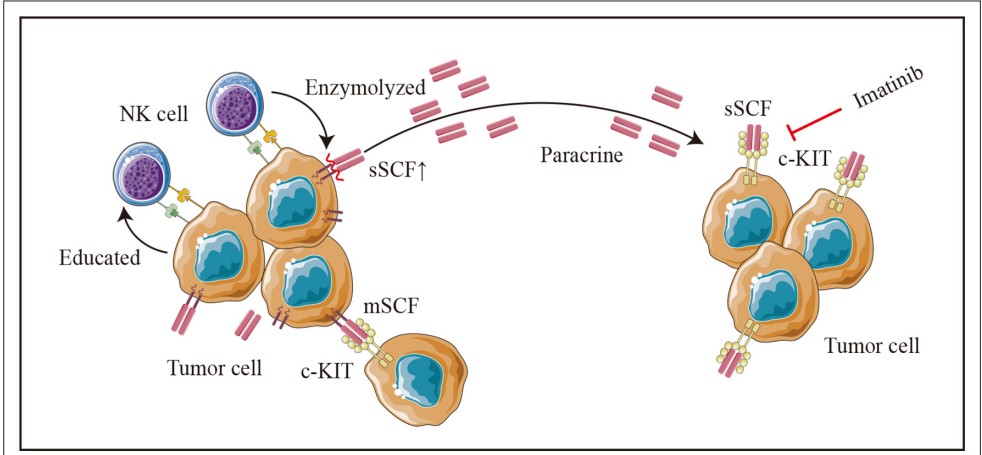

**Figure 9.** Schematic diagram. Colon cancer cells (HCT-116 cells) educate natural killer (NK) cells as resting status depends on cell-to-cell interaction. Tumor-educated NK cells subsequent enhances tumor malignancy in a paracrine manner by elevating tumor-derived sSCF.

soluble forms (sSCF) (*Hsu et al., 1997*), enhances the growth and migration of cancer cells such as cervical cancer (*Aguilar et al., 2014*), Ewing's sarcoma (*Landuzzi et al., 2000*), and CRC (*Yasuda et al., 2007*). Although the dimeric mSCF is remarkably more active and induces a more persistent activation of the receptor (*Hsu et al., 1997*; *Langley et al., 1993*), the tumor-promoting effect of the supernatant from the co-culture system in the current study and the reversion effect of imatinib for the co-culture supernatant induced pro-oncogenic effects suggests an important role of sSCF on colon cancer cells. However, no up-regulation of KITLG (the SCF encoding gene) was found in HCT-116 after co-culturing with NK cells (*Figure 8—figure supplement 1A*). In addition, when using DLD-1, a colon cancer cell with a nearly 15 times lower expression of KITLG than HCT-116 (*Figure 8—figure supplement 1B*), we also did not observe co-cultured supernatant induced tumor-promoting effects. We thus hypothesized that in our experimental system, NK cells may promote the cleavage and release of the extracellular part of SCF on the tumor cell surface after direct contact with tumor cells, and thus increase the concentration of sSCF, which induces the malignant phenotype of colon cancer cells in a paracrine manner (*Figure 9*). However, the detailed mechanism still needs to be further studied.

In summary, we have unveiled the spatial and cellular profiles of TME from tumors and paratumor tissues of CCLM. Our analysis uncovered the different states, functions, and dynamic nature of NK cells in different CCLM settings, which can be used for further identification of regulatory mechanisms and for developing potential therapeutic targets.

## Materials and methods

### Data sources

In this study, we integrated four independent public datasets that contained single-cell RNA-sequencing and ST data (10× Genomics) (derived from previously published research at the website: http://www.cancerdiversity.asia/scCCLM/) (*Wu et al., 2022*). And two bulk transcriptomics data of colon cancer (downloaded from TCGA cohort COAD at the website: https://xena.ucsc.edu/ and NCBI's Gene Expression Omnibus with accession number GSE29623).

### Single-cell RNA-sequencing analysis

The R (v4.0.5) package Seurat (v4.1.0) (*Hao et al., 2021*) was used to process the scRNA data. Since dataset quality control had been performed and Seurat objects had been created in previous studies, we did not further filter the scRNA-seq data or remove the batch effects. The SCTransform method was used to normalize the data. After selecting 2000 highly variable genes using the Find Variable Features function in Seurat, principal component analysis was performed using these genes to reduce the data dimensions. Based on the ElbowPlot function in Seurat, we chose the top 30 principal components to run the FindNeighbors function. Next, the cells were clustered using the FindClusters function with a resolution of 0.1 for clustering immune cells. A uniform manifold approximation and projection algorithm was used for data visualization, as previously reported (*Peng et al., 2023*). DEGs of each subset were identified using the FindAllMarkers function in Seurat. SingleR (v1.0.0) was used to name each cluster (*Zheng et al., 2021*). Additionally, the subgroup of NK cells was defined by the preferential expression marker of the resting and activated NK cells described in the CIBERSORT.

### Pseudotime analysis

To analyze the differentiations of NK cells, monocle2 (http://cole-trapnell-lab.github.io/monocle-release), which uses an algorithm to learn the changes in gene expression sequences that each cell must undergo as part of a dynamic biological process (*Hong et al., 2022*), was used for pseudotime trajectory analysis to identify the transitional relationships among different clusters. The cells were reduced dimensionally using the DDRTree method, sequenced in pseudotime, and finally visualized (*Qiu et al., 2017*).

### Bulk RNA-seq data analysis

The bulk transcriptome RNA-seq data and corresponding clinical data were obtained from The Cancer Genome Atlas of colon adenocarcinoma through the UCSC Xena browser (GDC hub) (https://gdc.xenahubs.net) (*Goldman et al., 2020*). In total, 459 colon cancer samples with survival information were downloaded, and 310 samples were finally enrolled (11 formalin-fixed samples were excluded,

and an additional 134 and 4 samples were excluded due to deletion of metastasis information and sequencing matrix, respectively). Transcriptomic data from 65 colon cancer samples in GSE29623 were obtained as the validation cohorts. CIBERSORT (*Chen et al., 2018*), xCELL (*Aran et al., 2017*), EPIC (*Racle et al., 2017*), and MCP-counter (*Becht et al., 2016*), which use gene expression to infer the proportions of tumor-infiltrating immune cells, were used to analyze the TME cell type. R packages survival (v3.2–10) and survminer (v0.4.9) were used for survival analysis. The Youden index was selected as the cutoff value to differentiate patients into distinct groups (high or low). The Kaplan–Meier survival curve was modeled using the survfit function. Subsequently, a Cox proportional hazards regression model was established to determine the independent risk factors. DEGs between the metastasis and non-metastasis groups were determined with the filtering condition of log2|fold change| > 1 and p-value <0.05, using the R package limma. GO and KEGG pathway functional enrichment analyses were performed using the clusterProfiler R package (v3.18.1) to assign various biological processes, molecular functions, cellular components, and pathways of identified marker genes in the cluster of interest (*Yu et al., 2012*), and p < 0.05 was regarded as statistically enriched. To explore the different KEGG pathways and hallmark gene sets between the metastasis and non-metastasis groups, GSEA was performed using data from The Molecular Signatures Database (c2.cp.kegg.v7.3.symbols) and the fgsea R package (v1.12.0) (*Liberzon et al., 2015*). Pathways with an adjusted p-value below 0.05 were deemed to be significantly enriched.

## ST data analysis

Seurat was also used for ST data processing and visualization. We used the SCT method to normalize the ST data; the functions SelectIntegrationFeatures, PrepSCTIntegration, FindIntegrationAnchors, and IntegrateData were used to integrate the ST data. An unsupervised clustering method was subsequently used to cluster similar ST spots. Cell population annotations were based on H&E staining sections and the highly variable genes in each cluster. Scores of cell-specific signatures (top 20 DEGs) from scRNA-seq were calculated using the AddModuleScore function. SpatialDimPlot and SpatialFeaturePlot were combined to visualize the cell expression level in the ST data (*Peng et al., 2023*). Since the tumor and non-tumor region was identified by immunohistochemistry as well as cell clusters in ST data, the differences of cell subgroup presence in tumor and non-tumor region were determined by visual inspection.

## Fluorescent multiplexed immunohistochemistry

We performed the fluorescent multiplexed immunohistochemistry by using the GZMK rabbit anti-human antibody (R&D; catalog no. MAB10216, 1:200), KIR2DL4 rabbit anti-human antibody (novus; catalog no. BNP2-17062, 1:600), CD56 rabbit anti-human antibody (ABSIN; catalog no. 171540, 1:200) following the manufacturer's instructions of four-color F-MIHC Kit (Absin, China). The expression of proteins of interest was evaluated as follows: GZMK, Absin 570 TSA Plus; KIR2DL4, Absin 520 TSA Plus; CD56, Absin 650 TSA Plus. The images were obtained by Case Viewer2.4 (3DHISTECH, Hungary).

## Cell lines and co-cultures

Human colon cancer cell lines, including HCT-116 and DLD-1 were purchased from the Cell Bank of the Chinese Academy of Sciences (Shanghai, China) and grown in a RPMI-1640 medium (Gibco) supplemented with 10% fetal bovine serum (FBS, Gibco). Human Foreskin Fibroblast cell lines, HFF-1, was also obtained from the Cell Bank of the Chinese Academy of Sciences (Shanghai, China) and grown in the Dulbecco's modified Eagle's medium (Gibco) containing 10% FBS. The human NK cell line NK-92 was purchased from the Cell Bank of the Chinese Academy of Sciences (Shanghai, China) and maintained in RPMI-1640 supplemented with 12.5% horse serum (Gibco), 12.5% FBS, 100 U/ml rhIL-2, 0.1 mmol/l b-mercaptoethanol, and 0.02 mmol/l folic acid. All the cells were cultured in a humidified chamber at 37°C and 5% $CO_2$. All the cells used in the study were short tandem repeat (STR) authenticated and tested negative for mycoplasma contamination.

Colon cancer cells were plated at a density of $1 \times 10^5$ cells/well in 6-well plates for 24 hr. Supernatants were collected and cells and debris therein were removed by centrifugation. NK cells were co-cultured with cancer cells in a ratio of 1:1 in a fresh mixed medium for a further 24 hr (CN group). Additionally, NK cells were also cultured in a mixed medium of supernatant and NK medium in the

ratio 1:1 (SN group) as well as fresh RPMI-1640 10% FBS and NK medium in the same ratio (MN group) for 24 hr (*Figure 7—figure supplement 5*). When indicated, co-cultures were performed in Transwell devices (JET biofil), maintaining the same ratios and culture times. Finally, as shown in *Figure 7—figure supplement 5*, the co-cultured supernatant of CN, SN, and MN group were collected, which determined as CNS, SNS, and MNS, to incubate fresh HCT-116 to explore the NK cell-mediated inductive effect on colon cells.

## Flow cytometry analysis

NK cells in each co-cultured group were collected and washed twice in 1× phosphate buffered saline (PBS). Then, they were incubated with APC-labeled CD56 (Biolegend), phycoerythrin (PE)-labeled KIR2DL4 (Biolegend), fluorescein isothiocyanate (FITC)-labeled GZMK (Biolegend), FITC-labeled CD9 (Biolegend), PE-labeled PD-1 (Biolegend), and PerCP-labeled CD49a antibody for 30 min, rewashed, and resuspended in PBS. FMO control was set up to define the position of the positive gate as shown in *Supplementary files 5 and 6*. BD flow cytometry was used for detection.

## CCK-8 cell viability and cell colony formation assay

Colon cancer cells (5000 cells/well) were cultured in 96-well plates with co-cultured supernatants from different co-culture groups alone or with both co-cultured supernatants and imatinib mesylate (2 μM) as previously reported (*Yasuda et al., 2007*) for 24 hr. At pre-determined time points, 10 μl of CCK-8 reagent (Dojindo, Japan) was added and incubated for 2 hr at 37°C, and then the absorbance was measured using a microplate reader (Thermo Scientific) at 450 nm. All experiments were carried out in triplicate.

For the cell colony formation assay, colon cancer cells (500 cells/well) were seeded in 12-well plates with co-cultured supernatants alone or with both co-cultured supernatants and imatinib mesylate at 37°C for 1 week. Then, cell colonies were fixed with 4% paraformaldehyde for 10 min and stained with 0.5% crystal violet for 5 min. Cell colonies containing >20 cells were counted. All experiments were carried out in triplicate.

## Transwell migration/invasion assays

The polycarbonate membrane in the transwell chambers was coated with Matrigel (Corning, NY, USA). Next, we transferred $1 \times 10^5$ cells from the serum-free medium with or without imatinib mesylate into the top chamber, added co-cultured supernatants alone or with both co-cultured supernatants and imatinib mesylate in the bottom chamber, and incubated at 37°C for 24 hr. Then, we removed the non-invading cells on the top side of the membrane by scrubbing, fixed the migrating or invading cells at the bottom side of the membrane with 4% paraformaldehyde, and stained with 0.5% crystal violet. The number of cells was counted under a microscope (Leica, London, UK) from four randomly chosen fields per well to determine the number of cells in each group.

## Total RNA extraction and quantitative real-time PCR

Total RNA was extracted from cells using TRIZOL reagent (Invitrogen, USA). RNA was reverse transcribed into cDNA using a PrimeScript RT Reagent Kit (Takara, Japan). Quantitative real-time PCR (qPCR) was performed using QuantStudio Test Development Software (Thermo Scientific, Waltham, MA, USA) with SYBR Green qPCR Master Mix (EZBioscience, Roseville, MN, USA). The sequence of KITLG and housekeeping gene GAPDH primers is listed in *Supplementary file 7*. KITLG mRNA data were normalized to that of GAPDH.

## Luminex liquid suspension chip detection and ELISA

Luminex liquid suspension chip detection was performed by Wayen Biotechnologies (Shanghai, China). The Bio-Plex Pro Human Chemokine Panel 48-plex kit was used in accordance with the manufacturer's instructions. Briefly, supernatants from different co-cultured groups were incubated in 96-well plates embedded with microbeads for 1 hr and then incubated with a detection antibody for 30 min. Subsequently, streptavidin-phycoerythrin (PE) was added in each well for 10 min, and values were read using the BioPlex MAGPIX System (Bio-Rad). The Human SCF ELISA kit (Solarbio, China) was used according to the manufacturer's instructions.

## Statistical analysis and visualization

All statistical analyses were performed using SPSS (v23.0; IBM SPSS, Chicago, IL, USA) and R (v4.0.5), and data visualization was performed on R packages Seurat (v4.1.0), ggplot2 (v3.3.5), ggsignif (v0.6.1), and pheatmap (v1.0.12).

## Acknowledgements

We thank Prof. Qiang Gao from Fudan University and Prof. Xiaoming Zhang from Shanghai Institute of Immunity and Infection, CAS for sharing the SC RNA-sequencing and ST data. We thank Zhejiang Key Laboratory of New Techniques for Diagnosis and Treatment of Critical Diseases of Pancreas and Liver and Zhejiang International science and Technology Cooperation base for tumor transformation research for supporting this study. We thank Editage Group (https://www.editage.cn/) for polishing the draft of this manuscript.

## Additional information

### Funding

| Funder | Grant reference number | Author |
|---|---|---|
| National Natural Science Foundation of China | 32070151 | Xian Shen |
| Key Research and Development Program of Zhejiang Province | 2020C03029 | Xiangyang Xue |
| Key Research and Development Program of Zhejiang Province | 2021C03120 | Xian Shen |
| Zhejiang Provincial Natural Science Foundation Project | Q24H160147 | Wangkai Xie |
| The Postdoctoral Fellowship Program of CPSF | GZC20231956 | Wangkai Xie |
| The Wenzhou Municipal Science and Technology Bureau Program | Y2020215 | Dan Xiang |
| Clinical Medicine Plus X - Scholars Project of the Second Affiliated Hospital of Wenzhou Medical University | | Mingdong Lu Xiangyang Xue |

The funders had no role in study design, data collection, and interpretation, or the decision to submit the work for publication.

### Author contributions

Chenchen Mao, Conceptualization, Data curation, Software, Formal analysis, Writing – original draft; Yanyu Chen, Conceptualization, Resources, Data curation, Software, Writing – original draft; Dong Xing, Conceptualization, Resources, Data curation, Investigation; Teming Zhang, Conceptualization, Resources, Data curation, Methodology, Writing – original draft; Yangxuan Lin, Software, Investigation; Cong Long, Conceptualization, Investigation; Jiaye Yu, Resources, Validation; Yunhui Luo, Methodology; Tao Ming, Validation; Wangkai Xie, Funding acquisition, Writing – review and editing; Zheng Han, Investigation, Methodology; Dianfeng Mei, Software; Dan Xiang, Funding acquisition, Validation, Writing – review and editing; Mingdong Lu, Supervision, Project administration, Writing – review and editing; Xian Shen, Xiangyang Xue, Supervision, Funding acquisition, Validation, Writing – review and editing

## Author ORCIDs
Yunhui Luo https://orcid.org/0009-0002-3914-5226
Xiangyang Xue https://orcid.org/0009-0008-3241-5282

Reviewer #1 (Public Review): https://doi.org/10.7554/eLife.97201.3.sa1
Reviewer #2 (Public Review): https://doi.org/10.7554/eLife.97201.3.sa2
Author response https://doi.org/10.7554/eLife.97201.3.sa3

## Additional files

### Supplementary files
- Supplementary file 1. Characterized genes for Single-cell transcriptomic analysis.
- Supplementary file 2. Characterized genes for spatial transcriptomic analysis.
- Supplementary file 3. Characterized genes of natural killer (NK) subsets.
- Supplementary file 4. Characterized genes of natural killer NK of different status.
- Supplementary file 5. Fluorescence Minus One control of CD56, CD9, PD-1 and CD49a.
- Supplementary file 6. Fluorescence Minus One control of KIR2DL4 and GZMK.
- Supplementary file 7. Primers for quantitative real-time PCR (qPCR).
- MDAR checklist
- Source code 1. Source code used in this study.

### Data availability
All data are available in the main text or the supplementary materials.

The following previously published datasets were used:

| Author(s) | Year | Dataset title | Dataset URL | Database and Identifier |
|---|---|---|---|---|
| Wu Y, Yang S, Ma J | 2022 | Spatiotemporal Immune Landscape of Colorectal Cancer Liver Metastasis at Single-Cell Level | https://www.biosino.org/node/project/detail/OEP001756 | National Omics Data Encyclopedia, OEP00001756 |
| Chen DT, Hernandez JM, Shibata D, McCarthy SM | 2014 | mRNA and microRNA profile in colon cancer | https://www.ncbi.nlm.nih.gov/geo/query/acc.cgi?acc=GSE29623 | NCBI Gene Expression Omnibus, GSE29623 |

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
