## [Editor Report · eLife assessment]

This **useful** study draws on published single-cell and spatial transcriptomic data of colon cancer liver metastasis to clarify the pro- and anti-tumorigenic properties of NK cells. The authors discover increased GZMK+ resting NK cells in the tumor tissue and reduced abundance of KIR2DL4+ activated NK cells. However, the evidence is currently **incomplete**, as the models used to validate the hypothesis and claims are not adequate and lack the necessary controls.

---

## [Referee Report · Reviewer #1 (Public Review)]

Summary:

Mao and colleagues re-analysed published spatial, bulk and single-cell transcriptomic datasets from primary colorectal cancers and colorectal cancer derived liver metastases. The analyses of paired cancer and non-cancer tissue samples showed that T cells are enriched in tumour tissue, accompanied by a reduction in the fraction of NK cells in the cancer tissue transcriptional datasets. Furthermore, the authors show that tumour tissue has higher fraction of GZMK+ (resting) NK cells and suggested a correlation between the presence of these cells and poor prognosis for cancer patients. In contrast, the increased frequency of KIR2DL4+ (activated) NK cells correlates with improved survival of cancer patients.

Strengths:

Authors performed a comprehensive analysis of published datasets, integrating spatial and single-cell transcriptomic data, which allowed them to discover enrichment of GZMK+ NK cells in cancer tissues.

Weakness:

The authors provided insufficient experimental evidence to support their claim that GZMK+ NK cells contribute to worse prognosis for cancer patients or promote cancer progression. While one can visually observe an increased fraction of GZMK+ NK cells compared to KIR2DL4+ NK cells in cancer tissues, no quantification is shown. They did not present any preclinical (animal model) or clinical data suggesting a causal relationship between NK cells and tumour growth. Thus, while a correlation may exist between the presence of GZMK+ NK cells and poorer tumour prognosis, causation cannot be claimed based on the available evidence. Furthermore, the in vitro data provided is limited to a single NK cell line derived from a lymphoma patient, which does not fully represent the diversity and functionality of human NK cells.

---

## [Referee Report · Reviewer #2 (Public Review)]

Summary:

This manuscript investigates the role of the abundant NK cells that are observed in colon cancer liver metastasis using sequencing and spatial approaches in an effort to clarify the pro and anti-tumourogenic properties of NK cells. This descriptive study characterizes different categories of NK cells in tumor and tumor adjacent tissues and some correlations. An attempt has been made using pseudotime trajectory analysis but no models around how these NK cells might be regulated is provided.

Strengths:

This study integrates multiomics data to attempt to resolve correlates of protection that might be useful in understanding NK cell diversity and activation. The authors have strengthened the study in revision by demonstrating the very strong correlation between Granzyme+ NK cells and the poor prognosis, but the main claims are only partially supported.

Weaknesses:

While this work is interesting, the power of such studies are in taking the discovered information and applying this to other cohorts to determine the strength and predictive power of the genes identified. It is also clear that these 'snapshots' analysed poorly take account of the dynamic temporal changes that occur within a tumour. It would have been good to see a proposed model of NK cell regulation as it might occur in the tumour (accounting for turnover and recruitment) beyond the static data. Further evidence linking mechanistic causality to prognostic outcome would provide significant data for approaches forward.

---

## [Author Response]

The following is the authors’ response to the original reviews.

**Reviewer#1:**
Comment #1: It is unclear how the fraction of NK cell populations is quantified in the spatial-seq datasets. Figures display spatial data with expression scores, but the method for calculating the score and determining NK cell presence in tumor tissue is ambiguous. Clarification is needed on whether the identification relied solely on visual inspection or if quantitative analyses using other criteria were conducted.

Thank you for your questions. We removed the background and made the accordingly modifications according to your demand. We used the AddModuleScore function in Seurat to quantify the main immune subpopulations in spatial-seq using the gene sets identified in single-cell-seq. Additionally, the tumor and non-tumor region was identified by immunohistochemistry as well as cell clusters in spatial-seq, it is rough that we can't quantify the NK cell presence in each region precisely. The consolation is that the differences of NK cell presence in tumor and non-tumor region is observable by visual inspection. The methodology has been supplemented in the revised manuscript (line 190-193).

Comment #2: The authors do not provide a clear definition of "resting" NK cells. It remains unclear whether they refer to a senescent state or a non-matured NK cell population. Furthermore, the criteria used to define resting and activated cells based on the expression of KIR2DL4, GPR183, GRP171, CD69, IFNG, GZMK, TTC38, CD160, and PLEKNF1 in Figure 4 are not well-defined. The expression patterns of these genes in Figure 4D are not distinct, and it is unclear which combination of genes was used to classify the populations. Clarification is needed on whether the presence of GZMK alone defines resting NK cells, or if the presence of any of the described genes (GZMK, TTC38, or CD160) is sufficient. Additionally, the method used for this classification, whether visual or algorithm-based, should be described.

Thank you for your question. The resting and activated NK cells was defined by the preferential expression of the described resting genes (AZU, BPI, CAMP, CD160,CD2, CDHR1, CEACAM8, DEFA4, ELANE, GFI1, GZMK, KLRC4, MGAM, MS4A3, NME8, PLEKHF1, TEP1, TRBC1, TTC38, ZNF135) and activated NK genes (APOBEC3G, APOL6, CCL4, CCND2, CD69, CDK6, CSF2, DPP4, FASLG, GPR171, GPR18, GRAP2, IFNG, KIR2DL4, KIR2DS4, LTA, LTB, NCR3, OSM, PTGER2, SOCS1, TNFSF14) in CIBERSORT. Actually, these marker genes were not specifically expressed in a single NK cells subset. On the other hand, combined with further flow cytometric analysis verification, the resting NK cell tend to be a decidual-like NK cells and tumor- infiltrated NK cells with higher expression of CD9, CD49a and PD-1.

Comment #3: Criteria used to define high or low NK cell presence/infiltration in Figure 5 are not described in the main text or figure legend. Since, the claim that the presence of the resting or activated NK cells predicts cancer prognosis is based on this figure, this needs to be clearly described.

Thank you for your questions. The activated and resting NK cell percentage in TCGA and GSE29623 was determined by CIBERSORT. Additionally, the infiltration of activated and resting NK cell was also determined by the AddModuleScore function using the gene sets of activated and resting NK cell identified in single-cell-seq, the differences of activated and resting NK cell presence in tumor and non-tumor region is also determined by visual inspection. We have amended in the main text and figure legend in the revised manuscript.

Comment #4: The absence of FMO controls for KIR2DL4 or GZMK and the lack of increase in GZMK expression during co-culture with tumour lines raises concerns since GZMK was used as a defining feature of resting NK cells.

Thank you for your questions. We did a new batch of flow experiments and FMO controls of all the markers used in the experiments were set up to define the precise positive gate locations.

**Author response image 1. sa3fig1:** The positive gate locations of CD56, GZMK, KIR2DL4, CD9, CD49a, PD-1 defined according to the FMO control.

Comment #5: All the co-cultures were performed with tumour cell line only and no healthy cells, such as human foreskin fibroblasts, were used as control. In the absence of a non-tumour cell line, it is very difficult to draw any conclusions. Furthermore, to claim that resting or activated NK cells are responsible for tumour migration or proliferation, it is important to at least isolate resting and activated NK cells ex vivo and culture with tumour lines, instead of NK cell lines.

Thank you for your questions. According to your suggestion, NK cells were co-cultured with human foreskin fibroblasts, the phenotype was identified by Flow cytometry. When co-cultured with HFF in direct contact (CN group), NK cells were also tending towards tissue infiltration state (high expression of CD9). However, the domestication effect is significantly reduced compared to co-culturing with tumor cells. Additionally, unlike supernatant of CNS group (NK and HCT were in contact) from NK and HCT co-culture system could significantly increase the migration of fresh HCT, fresh HCT underwent a limited increase (no statistical significance was found) in migration when cultured in the supernatant from the co-culture system in which NK and HFF were in contact (CNS group), but not when co-cultures were performed in the cell supernatant (SNS group) and fresh medium (MNS group). Finally, we tried to isolate resting and activated NK cells from fresh colon cancer surgical specimen. Unfortunately, the NK cells were too few to perform further functional experiments such as migration and proliferation.

**Author response image 2. sa3fig2:** Phenotype switch of NK cells in different co-cultured system and the corresponding NK cell-mediated effect on cell migration of fresh colon cancer cell (HCT-116). A-B: NK cells underwent phenotype switch (high expression of CD9) when cocultured with HCT and HFF, the phenotype switch was more obvious when co-cultured with HCT. CN: NK cells cocultured with HCT/HFF; SN: NK cells cocultured with supernatant of HCT/HFF; MN: NK cells cocultured in fresh medium. C-E: Transwell assay showed the only tumor co-cultured NK mediated the inductive effect on cell migration of colon cancer cell (HCT-116). CNS: Colon cancer cells were cultured in the supernatant from co-culture system that NK and HCT/HFF were cultured in direct contact; SNS: Colon cancer cells were cultured in the supernatant from co-culture system that NK cocultured with supernatant of HCT/HFF; MNS: Colon cancer cells were cultured in the fresh medium.

Comment #6: It seems that flow cytometric analyses and GZMK and KIR2DL4 staining were performed without cell permeabilization. Could authors confirm if this is accurate, or if they performed intracellular staining instead?

Thank you for your questions. For GZMK, which known as the secretory protein, flow cytometric analyses were performed both with (Fig.3) and without cell fixation and permeabilization, no significant differences were found among each group. The difference is that GZMK was nearly all negative without fixation and permeabilization while it is all positive with fixation and permeabilization. Conditions of flow cytometry analyses for GZMK may need further optimization or GZMK may not be a suitable flow cytometric marker for resting NK cells. On the other hand, for membrane protein such as CD56, CD9, CD49a, KIR2DL4, PD-1, staining was performed without cell permeabilization.

**Author response image 3. sa3fig3:** Phenotype switch (CD56+, GZMK+) of NK cells was analyzed by FACS after fixation and permeabilization in different co-cultured groups. CN: NK cells cocultured with colon cancer cells; SN: NK cells cocultured with supernatant of cancer cells; MN: NK cells cocultured in fresh medium.

Comment #7: The identity of the published datasets used for analysis is not provided, and references are not cited in the results section.

Thank you for your questions. We are sorry for the neglect of our previous work. We have added the information in the revised manuscript (section of Materials and Methods) (Line 123-128).

Comment #8: References are difficult to locate, as the main text follows APA style while the reference section is organized numerically with no clear order.

Thank you for your questions. We have modified the format of the references in the revised manuscript.

Comment #9: Figure 3 shows volcano plots showing DEG genes between tumor and healthy tissue NK cells are not described clearly, and authors did not discuss the significance of these genes, highlighted in the plot.

Thank you for your questions. Volcano plots of Figure 3 showed the DEGs between colon cancer with metastasis and without metastasis in TCGA database. We focused on the genes which were enriched in the pathway of “Natural killer cell mediated cytotoxicity” and found nearly all the genes enriched in the pathway were down-regulated in the colon cancer with metastasis. We have modified the description in the result section and added the description of importance of these genes in the discussion section in the revise manuscript (Line 322-326).

Comment #10: The meaning of "M0" and "M1" in Figures 5A and 5B is unclear and should be defined in the text.

Thank you for your questions. "M0" and "M1" in Figure 5A and 5B means “colon cancer without metastasis” and “colon cancer with metastasis”, respectively. We have modified in the revise manuscript (Line 350-354).

Comment #11: Terms such as "dynamic remodelling of NK cells" and "landscape of NK cells" are used without explanation, necessitating clarification of their meaning.

Thank you for your questions. We have modified in the revise manuscript (Line 331-334).

Comment #12: In vitro assays are described vaguely, making it difficult for readers to understand. More clarity is needed in describing these assays.

Thank you for your questions. We have added clarification in the revise manuscript (Line 205-211).

**Reviewer #2:**
Comment #1: This manuscript investigates the role of the abundant NK cells that are observed in colon cancer liver metastasis using sequencing and spatial approaches in an effort to clarify the pro and anti-tumorigenic properties of NK cells. This descriptive study characterises different categories of NK cells in tumor and tumor-adjacent tissues and some correlations. An attempt has been made using pseudotime trajectory analysis but no models around how these NK cells might be regulated are provided.

Thank you for your questions. The single-cell sequencing data enrolled in this study are CD45 positive immune cells and do not involve tumor cells, cellular communication analysis between NK cells and tumor cells cannot be conducted. The change process of NK can only be predicted through pseudotime trajectory analysis. Our hypothesis is that tumor cells domesticate NK cells into a tumor- infiltrated NK cells through direct contact, and flow cytometry experiments have also confirmed that tumor cells can only have such domestication through direct contact with NK cells (with prominent high expression of CD9). However, the detailed mechanism remained unclear.

Comment #2: A small number of patients are analyzed in this study. The descriptive gene markers, while interesting, need to be further validated to understand how strong this analysis might be and its potential application.

Thank you for your questions. The sample size included in this study is indeed a bit small, which is also a limitation of our study. However, this is the only large sample single-cell sequencing dataset could be found that includes primary colon cancer tissues, paired paratumor normal colon tissues, paired liver metastatic cancer tissue, and paired paratumor normal liver tissues. We will expand the sample size to further verify the current conclusion in subsequent experiments. In addition, the marker genes of different NK groups used in this study refer to the CIBERSORT's classification of activated NK cells and resting NK cells, which is a widely recognized indicator. We will verify the expression and clinical application value of the screened genes in tissues in subsequent studies.

Comment #3: Figure 1C and other figures throughout the paper. It is not clear how marker genes were selected.

Thank you for your questions. The marker genes displayed in the Figure.3C were the highly variable genes of each cell group as well as the marker genes of each immune cells, such as T cells (CD3D, CD3E), NK cells (NKG7, KLRD1), monocytes (LYZ, S100A8, S100A9), B cells (CD79A), plasma cells (JCHAIN, IGHA1, IGHA2), Neutrophils (CXCL8, FCGR3B).

Comment #4: Figure 1E. P and T have not been defined. Lines should not connect the datasets as they are independent assessments.

Thank you for your questions. P and T means paratumor normal tissues and tumor tissues, respectively. Which have been added in the caption of Figure 1E. Additionally, the single cell sequencing samples included in the study were paired, with primary colon cancer tissues, paired normal tissues adjacent to colon cancer, paired liver metastatic cancer tissue, and paired normal liver tissues from 20 colon cancer patients with liver metastasis, paired test analysis was thus performed.

Comment #5: Figure 2C. It is unclear what ST-P1 means. This is not a particularly informative figure.

Thank you for your questions. We are sorry that it was our annotation error. Actually, it is the spatial transcriptome of the primary colon cancer tissue and liver metastasis tissue of four patients. We have made the modifications in the revised manuscript.

Comment #6: Multiple figures - abbreviations are used but not provided in the legend. They occur in the text but are not directly related to the figures where they are used to label axes or groups.

Thank you for your questions. We have rechecked and made corresponding modifications in the revised manuscript.

Comment #6: Patients: it is not clear what other drugs patients have been exposed to or basic data (sex, age, underlying conditions etc)

Thank you for your questions. The baseline data of the patient of SC dataset and ST dataset were showed in the Table.1 and Table.2 followed, respectively. They were not presented before as no patients characteristics related analysis was performed in the current study.

**Author response table 1. sa3table1:** The baseline data of patient from single cell sequencing database.

Patient	Age	Sex	TNM	Neoadjuvant chemotherapy
P1	71	Male	T1N0M1	N
P2	78	Female	T3N1M1	N
P3	60	Female	T2N0M1	N
P4	26	Female	T3N1M1	N
P5	64	Male	T2N0M1	Y (PR, FOLFOX)
P6	71	Male	T3N1M1	N
P7	52	Female	T3N1M1	Y (PR, XELOX)
P8	53	Male	T3N0M1	N
P9	35	Male	T1N0M1	Y (PR, XELOX)
P10	66	Male	T3N1M1	N
P11	43	Female	T3N1M1	N
P12	68	Male	T3N1M1	N
P13	69	Female	T3N1M1	Y (PD, XELOX)
P14	46	Female	T3N1M1	Y (PD, XELOX)
P15	51	Female	T3N1M1	Y (PR, FOLFOX)
P16	64	Female	T3N1M1	N
P17	49	Female	T3N1M1	N
P18	55	Male	T3N1M1	Y (PD, XELOX)
P19	63	Female	T2N0M1	Y (PD, XELOX)
P20	54	Male	T3N0M1	Y (SD, FOLFOX)

**Author response table 2. sa3table2:** The baseline data of patient from spatial transcriptome database.

Patient	Age	Sex	TNM	Treatment
ST-P1	73	Male	T3N0M1	N
ST-P2	67	Female	T3N1M1	N
ST-P3	72	Male	T3N1M1	Y (XELOX)
ST-P4	68	Female	T3N1M1	Y (XELOX)